# Impact Velocity Measurement Method Based on Trajectory and Impact Position

**DOI:** 10.3390/s22218288

**Published:** 2022-10-28

**Authors:** Hui Liu, Jingfan Wang, Yuantao Wu

**Affiliations:** 1School of Automation, Xi’an University of Posts & Telecommunications, Xi’an 710121, China; 2Shaanxi Institute of Metrology Science, Xi’an 710100, China

**Keywords:** impact velocity, low-speed sensor, falling trajectory, impact position

## Abstract

The impact velocity of falling weight is an instantaneous quantity. Currently, measurement of impact velocity relies on high-speed sensors to capture the moment of impact. The trajectory-position measurement method (TPMM) is proposed in this study. The main steps are: (1) The impact position is used to capture the impact time. It can be measured when the falling weight is stationary. (2) The discrete falling trajectory is measured and a new empirical regression algorithm is proposed to fit the expression of falling trajectory. (3) The impact velocity is obtained by taking the impact time into the first derivative of the trajectory expression. For 1–5 m falling height, the simulation shows that the relative maximum error and relative expanded uncertainty of the proposed method are less than 0.481% and 0.442%, respectively. Then, the actual experiment is carried out to verify the simulation. The proposed method has high accuracy and low uncertainty. The reasons are: (1) Only a low-speed displacement sensor is need for impact velocity measurement. It is easier to improve accuracy and stability of a low-speed sensor. (2) The empirical regression algorithm can improve the stability of falling trajectory fitting.

## 1. Introduction

An impact test is often used to test the mechanical properties of geotechnical soil and asphalt, or the quality of helmets [1,2]. The falling weight impact testing machine is the most commonly used impact test equipment. The falling weight achieves impact by free falling. Impact velocity is necessary to calculate the impact force and needs to be accurately measured. China released the ‘Calibration Specification for Falling weight impact testing machines’ (JJF 1445–2014) in 2014 [3]. The maximum permissible error (MPE) of the impact velocity measurement needs to be less than 0.5%.

Currently, measuring the time while a falling weight passes a fixed distance is the most commonly used impact velocity measurement method [4,5], and its principle is shown in Figure 1.

In Figure 1, two position detection sensors are placed at the impact position. Position detection sensors are used to record the time when the falling weight sweeps over the sensor. Position 1 and Position 2 are the positions of the falling weight detected by sensor one and sensor two, respectively. The moment is recorded when the sensor detects the falling weight. Furthermore, the time of the falling weight passing over two position detection sensors can be calculated. Finally, the impact velocity can be calculated according to Equation (1):(1)$$v=\frac{S}{t1-t0}$$
where S is the distance between two position detection sensors. t1,t0 are the moment recorded by the two sensors. If the distance S is small, the average velocity v is approximately equal to the impact velocity.

The above method requires the sensor to have a high response-speed to capture the falling hammer at the moment of impact. N. J. Fang [6] calculated the measurement uncertainty of this method. Regarding Fang’s approach, the measurement uncertainties under different impact velocities were calculated. The results show that the measurement uncertainty would increase with the increase of the impact velocity because the falling weight is more and more difficult to capture.

Among other velocity measurement methods, particle image velocity (PIV) and laser Doppler velocity (LDV) are commonly used. PIV uses CCD or CMOS to collect the image information of the object. The image segmentation algorithm is then used to obtain the position information of the object. Finally, the velocity can be calculated by combining the distance and time difference of the falling weight between the two images [7]. T. B. Dinh [8] used the PIV method to measure the velocity of bubbles in the fluid. W. C. Liu [9] measured the flow velocity of the river using the PIV method. The most similar areas in two successive images were identified. The distance and time of river flow was then obtained to calculate the flow velocity. The LDV method measures velocity based on the Doppler effect of the laser. The frequency of the laser changes when it is received by a moving object. The velocity can then be obtained by measuring the frequency shift [10]. M. M. Xu [11] measured the relative velocity of two satellites using the LDV method.

There is an essential difference between the above research and impact velocity measurement. The impact velocity measurement needs to measure the velocity at a specific time. And the PIV and LDV methods are mainly used to measure the velocity without special time requirements. The PIV and LDV methods also have few studies on the velocity measurement at a specific time. To capture the velocity when static friction becomes sliding friction, B. Denkena [12] used a 10,000~30,000 fps high-speed camera to photograph the friction contact surface, and the image was processed using PIVlab to obtain the velocity [12]. C. Lee [13] used a 20 KHz line scan camera to measure the impact velocity of a falling weight. Y. Fujii [14] measured the impact velocity of a heavy hammer with the LDV method, and the sampling frequency was 2000 times per second.

Including measurement methods based on position detection sensors, the above methods all need to capture the velocity at the impact moment of the falling weight by the response speed of the sensor, which is a massive challenge for the measurement system.

The impact velocity can be calculated by measuring the falling height if the air resistance is ignored, but this will cause large error [15]. If an accurate falling trajectory is obtained, the measurement error caused by air resistance can be corrected. The falling height is easy to measure with an ordinary low-speed sensor while the falling weight is stationary at the impact position. Based on this, this study proposes a measurement method that can accurately capture impact velocity without high-speed sampling. The laser displacement sensor is used to measure and fit the falling trajectory of the falling weight. After that, the accurate impact time is derived by combining the impact position and the falling trajectory. Finally, the impact velocity can be obtained by bringing impact time into the first derivative of the falling trajectory. We call the proposed method the trajectory-position measurement method (TPMM for short). Through simulation and actual experiment, the impact velocity can be measured accurately and stably by a low-speed sensor.

## 2. Principle of Trajectory-Position Measurement Method (TPMM)

As shown in Figure 2, a laser displacement sensor is fixed above the falling weight, and the laser is emitted in the direction of gravity, that is, the same direction as falling.

As shown in Figure 3, the specific measurement steps are as follows:

1.Impact position measurement:

The impact position, denoted by $s_{impact}$, can be measured while the falling weight is stationary at the impact position. The measurement can be carried out when the falling weight is stationary after falling, or the falling weight is placed on the impact position in advance. The first method was used in this study.

2.Falling trajectory measurement:

The laser displacement sensor is used to measure the position of the falling weight. The discrete sampling data is denoted by ${\left(s\left(t_{0}\right),\ldots ,s\left(t_{N-1}\right)\right)}^{T}$. $t_{0},\ldots ,t_{N-1}$ is the sampling time. N is the length of sampling data.

3.Data identification of free-falling stage:

According to the sampling data, the start time and the impact time of falling are preliminarily determined.

Start time: Before falling, the starting position is measured more than ten times, and the standard deviation is calculated, denoted by starting standard deviation. After that, if the standard deviation of five consecutive measured data is larger than 3 × stable standard deviation, the last measurement datum is considered as the start time.

Impact time (preliminarily determined): After the start time is determined, the standard deviation of five consecutive measured data is continuously calculated. If the standard deviation starts to decrease, the sampling time of the first of the current standard deviation calculation datum is the impact time. For the impact test, only the velocity at the first impact is concerned. Therefore, the subsequent impact caused by rebound is no longer concerned.

4.Falling trajectory expression fitting:

The expression of the falling trajectory is fitted through the data of the falling stage. To improve the accuracy of fitting, the force in the falling process is analyzed to determine that the third-order polynomial is used for fitting. In addition, the fitting objective function contains the regular term obtained through theoretical analysis. The details are as follows:

Determination of order of falling trajectory:

The falling weight is subjected to gravity and frictional resistance during falling. The frictional resistance is proportional to the square of the falling velocity [16]. However, friction resistance and falling velocity are approximately linear in correlation at low-speed. The force F of the falling weight during falling is shown in Equation (2):(2)F=mg−kv
where m is the mass of the falling weight, g is the acceleration of gravity, k is the air resistance coefficient in linear approximation, and v is the falling velocity.

The motion equation of the falling weight during falling is shown in Equation (3):(3)$$\frac{F}{m}+\frac{kv}{m}=g\Rightarrow \frac{d^{2}s}{dt^{2}}+\frac{k}{m}\cdot \frac{ds}{dt}=g$$
where s is the position of the falling weight.

According to reality, the initial value of Equation (3) is as follows.

Initial velocity is 0 $\Rightarrow {\frac{ds}{dt}|}_{t=0}=0$.

Initial position is 0 ⇒s(0)=0.

Initial acceleration is g $\Rightarrow {\frac{d^{2}s}{dt^{2}}|}_{t=0}=g$.

The analytical solution of Equation (3) is shown in Equation (4):(4)$$s=\frac{gt}{k}-\frac{g}{k^{2}}+\frac{ge^{-kt}}{k^{2}}$$

Taylor expansion is carried out for Equation (4) at t=0, as shown in Equation (5): (5)$$s=\overset{\infty }{\underset{i=2}{\sum }}\left(\frac{gk^{i-2}t^{i}}{i!}\right){\left(-1\right)}^{i}$$

According to Equation (5), when i>3, the order of k exceeds 1, and *k* is far less than 1, so the term after i>3 will decay rapidly and be ignored. Therefore, 3-order polynomial fitting is used to fit the falling trajectory of the hammer, as shown in Equation (6).
(6)$$s\left(t\right)=a_{3}t^{3}+a_{2}t^{2}+a_{1}t+a_{0}$$

Fitting objective function and solving method:

The multiple regression algorithm can be used to fit the measured data extracted in step (3) to obtain $a_{3},a_{2},a_{1},a_{0}$. In addition, according to the above analysis, the theoretical approximate value of $a_{3},a_{2},a_{1},a_{0}$ can be determined. $a_{2},a_{1},a_{0}$ reflect the initial acceleration, velocity, and position, respectively. Therefore, the theoretical approximations of $a_{2},a_{1},a_{0}$ are $\frac{g}{2}$, 0, 0, respectively. $a_{3}$ reflects the influence of resistance. Because the velocity is low, the value of $a_{3}$ is approximately around 0. If the theoretical approximate value is added to the objective function of multiple regression as a regular term, the influence of the measurement data error on the regression results will be reduced [17]. The objective function is shown in Equation (7). In this study, this algorithm is called empirical regression.
(7)$$\Gamma =\overset{N-1}{\underset{j=0}{\sum }}{\left(s\left(t_{j}\right)-a_{3}t_{j}{}^{3}-a_{2}t_{j}{}^{2}-a_{1}t_{j}-a_{0}\right)}^{2}+\eta \left(a_{3}{}^{2}+{\left(a_{2}-\frac{g}{2}\right)}^{2}+a_{1}{}^{2}+a_{0}{}^{2}\right)$$
where η Is the weight of the regular term. According to previous research [18], the value of η should be small and does not need to be accurate, 0.1 is taken in this study. The role of η is explained in detail in the discussion session.

We solved the minimum value of Equation (7) to get the values of $a_{3},a_{2},a_{1},a_{0}$, as shown in Equation (8).
(8)$$\{\begin{array}{c} \frac{\partial \Gamma }{\partial a_{3}}=0\\ \begin{array}{c} \frac{\partial \Gamma }{\partial a_{2}}=0\\ \frac{\partial \Gamma }{\partial a_{1}}=0 \end{array}\\ \frac{\partial \Gamma }{\partial a_{0}}=0 \end{array}$$

5.Impact time calculation:

Make
(9)$$s_{impact}=s\left(t\right)=a_{3}t^{3}+a_{2}t^{2}+a_{1}t+a_{0}$$

Then, the accurate impact time, denoted by $t_{impact}$ can obtained by solving Equation (9).

6.Impact velocity calculation:

Impact velocity, denoted by $v_{impact}$ can be obtained by bringing $t_{impact}$ into $s^{\prime }\left(t\right)$. As per Equation (10):(10)$$v_{impact}=s^{\prime }\left(t_{impact}\right)=3a_{3}t_{impact}{}^{2}+2a_{2}t_{impact}+a_{1}$$

## 3. Experimental Verification

Simulation and actual experiments are used to verify the trajectory-position measurement method. The simulation tests the accuracy and uncertainty of the proposed method at different falling heights. The actual experiment tests the measurement uncertainty at the height of 905 mm.

### 3.1. Simulation Experiment

In Figure 4, the specific steps of the experiment are as follows:

1.Falling trajectory simulation:

Set the falling start time as $t_{0}=0$, the falling height as $s_{impact}$, the impact time as $t_{impact}$, and the total simulation time as $2t_{impact}$. Set the time interval of simulation data as dt. The time corresponding to each simulation data is $t_{j},j=0,1,\ldots ,N^{*}-1(t_{0}\leq t_{j}<2t_{impact},t_{j+1}-t_{j}=dt)$. $N^{*}$ is the length of simulation data. Set $s\left(t_{0}\right)=0$, $v\left(t_{0}\right)=0$, $a\left(t_{0}\right)=g$.

From time $t_{1}$, perform the following steps at each simulation time:

Calculate current position.
(11)$$s\left(t_{j}\right)=s\left(t_{j-1}\right)+v\left(t_{j-1}\right)dt$$

Calculate current velocity.
(12)$$v\left(t_{j}\right)=v\left(t_{j-1}\right)+a\left(t_{j-1}\right)dt$$

Calculate current acceleration.
(13)$$a\left(t_{j}\right)=\{\begin{array}{c} g-\frac{v{\left(t_{j}\right)}^{2}C\rho A}{2M},v\left(t_{j}\right)\;\mathrm{downword}\\ g+\frac{v{\left(t_{j}\right)}^{2}C\rho A}{2M},v\left(t_{j}\right)\;\mathrm{upword} \end{array}$$
where C is the air resistance coefficient, which is between 0.5 and 1. ρ is the air density, A is the area under resistance, and M is the object’s mass.

In addition, when the falling height is larger than the $s_{impact}$, the velocity will reverse and decrease proportionally to simulate rebound. 

2.Falling trajectory sampling:

According to the actual performance of the sensor, the fall trajectory simulation data is sampled. The sampling interval is set to $dt_{s}\gg dt$, and the sampling data is added with a uniformly distributed random error. Here, the reason for adding random error is explained. Error is divided into systematic error and random error. Systematic error refers to stable error, which can be reduced by compensation even if it is nonlinear. On the other hand, the random error is challenging to deal with because it reflects the instability of the sensor and external uncertain interference. Therefore, all errors are considered as random errors during simulation. This is equivalent to simulating while the sensor is in the worst state. The simulation results have higher reliability in the actual environment.

3.Impact velocity calculation and comparison:

The impact velocity is calculated according to the sampling data and compared with the impact velocity calculated from the simulation data to evaluate the measurement accuracy. The methods involved in comparison are as follows:


(1)TPMM.(2)A key point of this study is the empirical regression algorithm. In order to verify the effect of the new algorithm, the empirical regression algorithm in TPMM is replaced by ordinary multiple regression for comparison. Specifically, replace the objective function in Equation (7) with Equation (14):
(14)$$\Gamma =\overset{N-1}{\underset{j=0}{\sum }}{\left(s\left(t_{j}\right)-a_{3}t_{j}{}^{3}-a_{2}t_{j}{}^{2}-a_{1}t_{j}-a_{0}\right)}^{2}$$This method is abbreviated as TPMM-N. (3)Distance and time measurement method (DTMM for short). This is the most commonly used method at present. DTMM needs to measure the position of falling weight before impact twice and calculate the distance and corresponding time. Then, the average velocity is the distance divided by time. When the distance is small and the measured position is close to the impact position, the average velocity can be approximately considered as the impact velocity. The falling weight position near the impact is included in the measurement data of the falling trajectory. Therefore, the impact velocity can also be measured by the DTMM. The specific methods are as follows: The velocity at each measurement time can be obtained by the first-order difference of the fall trajectory. Since the maximum velocity occurs at the impact time, the impact velocity is the maximum value of the first-order difference of the falling trajectory, as shown in the Equation (15):
(15)$$v_{impact}=\underset{j=1\ldots N-1}{\max}\frac{s\left(t_{j}\right)-s\left(t_{j-1}\right)}{t_{j}-t_{j-1}}$$
where $s\left(t_{j}\right)$ and $s\left(t_{j-1}\right)$ are the position of the falling weight at $t_{j}$ and $t_{j-1}$, respectively.


In addition, for the sensors used in the actual experiment, although the nominal error of the sensor is ±5 mm, part of the error is systematic error, and the random error is less than ±5 mm. DTMM only cares about two close positions near the impact, and part of the systematic error is offset when calculating the distance in Equation (15). Therefore, the accuracy of the sensor is higher for DTMM. During DTMM simulation, the error of the sensor is set to ± 1 mm. 

The values of simulation parameters are shown in Table 1.

Heights of 1, 2, 3, 4, and 5 m were simulated, and the falling trajectory is shown in Figure 5.

Each height was simulated 100 times, and the sampling error of each simulation is different. The measurement results are shown in Figure 6.

According to 100 measurement results at each height, the relative maximum error and relative expanded uncertainty are calculated, as shown in Equations (16) and (17), respectively, and the true value of velocity comes from simulation data.
(16)$$\mathrm{relative}\;\mathrm{maximum}\;\mathrm{error}=\frac{\underset{i=1\ldots 100}{\max}\left(v_{impact}{}_{i}-v_{impact}{}_{r}\right)}{v_{impact}{}_{r}}\times 100\%$$
(17)$$\mathrm{relative}\;\mathrm{expanded}\;\mathrm{uncertainty}=2\times \frac{\mathrm{std}\left(v_{impact}{}_{i}-v_{impact}{}_{r}\right)}{v_{impact}{}_{r}}\times 100\%\;\;$$
where $v_{impact}{}_{i}$ is the result of the *i*th measurement, $v_{impact}{}_{r}$ is the true value, $\mathrm{std}\left(v_{impact}{}_{i}-v_{impact}{}_{r}\right)$ is the standard deviation of 100 measurements.

In addition, according to reference [6], the relative expanded uncertainty of the most commonly used measurement method based on position detection sensors is calculated for comparison. The results are shown in Table 2.

### 3.2. Actual Experiment

The specific steps of the experiment are as follows:

Actual experience tests the proposed method by relative expanded uncertainty (Appendix A). As shown in Figure 7, the experimental device includes a laser displacement sensor, falling weight, guide rail, data acquisition system (Ni DAQ), power supply, and a computer. The falling weight is cylindrical, with a hole in the middle. The laser emission direction of the laser displacement sensor is the same as the falling direction. The hole diameter is 2 mm bigger than the guide rail. The material is rubber. The parameter values of the actual experiment are shown in Table 3.

Ten batches of actual measurement experiments were carried out. TPMM, TPMM-N, and DTMM are used to calculate the impact velocity according to the measurement data, respectively. The standard deviation and mean value are used to calculate the relative expanded uncertainty of the measurement results of different methods, as shown in Equation (18):(18)$$\mathrm{Relative}\;\mathrm{expanded}\;\mathrm{uncertainty}=2\times \frac{\underset{b=1\ldots 10}{\mathrm{std}}\left(v_{impact}{}_{b}\right)}{\underset{b=1\ldots 10}{\mathrm{mean}}\left(v_{impact}{}_{b}\right)}\times 100\%\;\;$$
where $v_{impact}{}_{b}$ is the impact velocity measurement result of batch b, and std and mean are the standard deviation and mean value, respectively. The results are shown in Table 4. The last line of Table 4 is the relative expanded uncertainty.

## 4. Discussion

In Table 4, the relative expanded uncertainty is basically consistent with the simulation experiment. The relative expanded uncertainty of TPMM is the lowest. Since there is no true value, the relative maximum error cannot be calculated. However, the uncertainty is sufficient to verify the proposed method. The minimal measurement uncertainty means that the TPMM can significantly reduce the uncertainty in the measurement and capture the impact velocity stably. In other words, TPMM solves the most challenging random error problem in impact velocity measurement based on ordinary low-speed sensors. The systematic error problem is easy to solve. One high-speed sensor can be used to calibrate and compensate multiple low-speed measurement devices. Therefore, low uncertainty is a performance concerned by users. The reasons are explained here.

1.The capture of impact velocity does not depend on the response speed of the sensor:

The impact velocity is an instantaneous quantity, and the current method requires the sensor to have a high-speed to improve the capture accuracy. However, under the premise of high response speed, it is challenging to require the sensor to maintain high measurement accuracy and stability simultaneously. The TPMM in this study captures impact velocity by impact position. As shown in Figure 8, during the measurement process, the whole falling stage can be divided into the initial stage, the falling stage, the rebound stage, and the stationary stage.

In the stationary stage, the falling weight is stationary at the impact position. The sensor has sufficient time to measure the impact position and obtain a lot of measurement data. The measurement accuracy can then be improved, and the measurement uncertainty can be reduced by averaging. 

2.The empirical regression can improve the fitting stability of falling trajectory:

In addition to the impact location, another key is the fitting of the falling trajectory. The regular terms of empirical regression can improve the stability of fitting results. According to Equation (7), the empirical regression algorithm is similar to ridge regression [17]. Both regression algorithms contain regular items. In Figure 9, as the weight of the regular term (η in (6)) increases, the distribution of the fitted coefficient changes from a ‘wide-low’ type to a ‘narrow-high’ type, indicating that the variance is constantly decreasing. Meanwhile, the expectation of a fitted coefficient tends to the value set by the regular term. For the ridge regression, the regular term is set to 0. The fitted coefficient then gradually deviates from the true value while the expectation tends to 0, and the accuracy gradually decreases. For the empirical regression, the regular term is set to theoretical approximation that is very close to the true value. So, the accuracy loss can be significantly reduced while the variance is reduced. Thus, the uncertainty can be greatly reduced while maintaining accuracy.

The measurement results of DTMM are obviously large, and the reasons are explained here:(1)The optical of the laser displacement sensor is not vertical, resulting in a larger distance measurement.(2)The nonlinear error of the sensor is larger than expected. The sampling interval of the sensor is 0.01 S, the two positions before the impact time are about 900 and 857 mm. We have calibrated the sensor and measured the error of at 903 and 848 mm, which are +1 and −2 mm, respectively (Appendix B). This also results in a larger distance measurement.(3)The above reasons all result in a large distance measurement. Meanwhile, the sampling interval of the sensor is 0.01 S. The distance errors are amplified in the velocity. For example, 1 mm distance measurement error causes a 0.1 m/s velocity measurement error. Therefore, the velocity measurement result of DTMM is large.

## 5. Conclusions

In this study, a new method for measuring falling weight impact velocity was proposed, called the TPMM. The laser displacement sensor was used to measure the falling trajectory and impact position of the falling weight to calculate the impact velocity. The characteristics of the proposed method are as follows:The capture of impact velocity does not depend on the response speed of the sensor, so the impact velocity can be measured using a general low-speed sensor. Low-speed sensors improve accuracy and stability more easily.A new empirical regression algorithm was proposed, which can integrate the theoretical analysis into the measurement results and improve the measurement accuracy and stability.

Through simulation and actual experiments, the proposed method has high measurement accuracy and small measurement uncertainty. The relative maximum error and expanded uncertainty are less than 0.5% for 1–5 m falling height. 

## Figures and Tables

**Figure 1 sensors-22-08288-f001:**
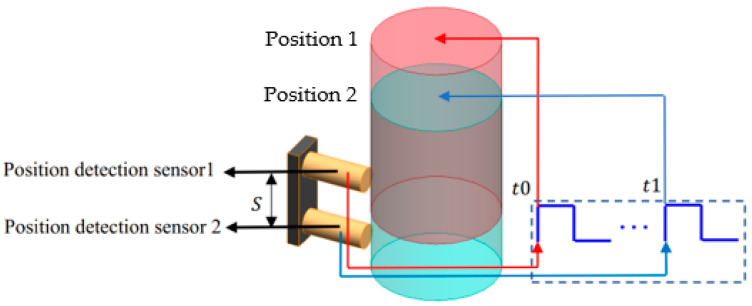
Impact velocity measurement principle based on position detection sensors.

**Figure 2 sensors-22-08288-f002:**
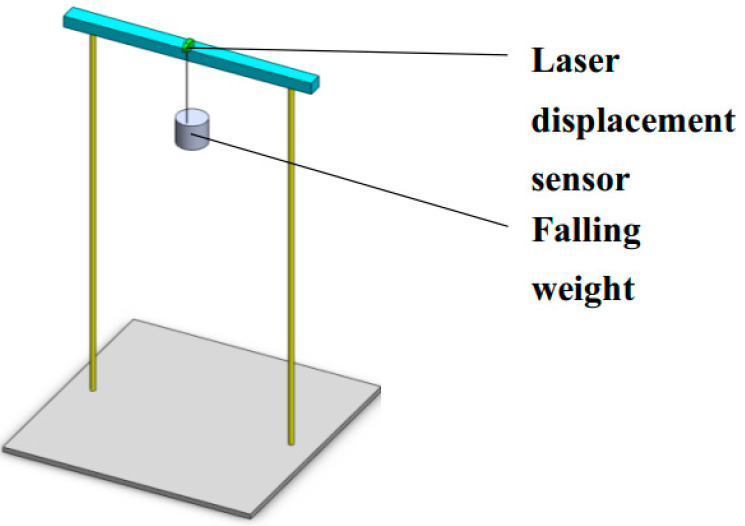
Hardware principle of TPMM.

**Figure 3 sensors-22-08288-f003:**
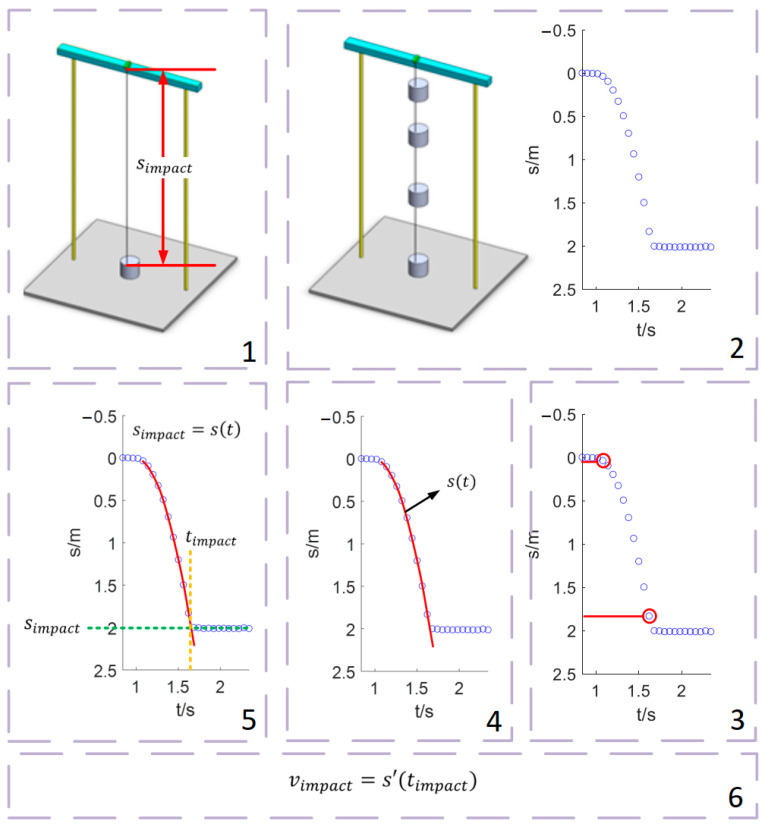
Specific steps of TPMM. 1. Impact position measurement; 2. Falling trajectory measurement; 3. Data identification of free-falling stage; 4. Falling trajectory expression fitting; 5. Impact time calculation; 6. Impact velocity calculation.

**Figure 4 sensors-22-08288-f004:**
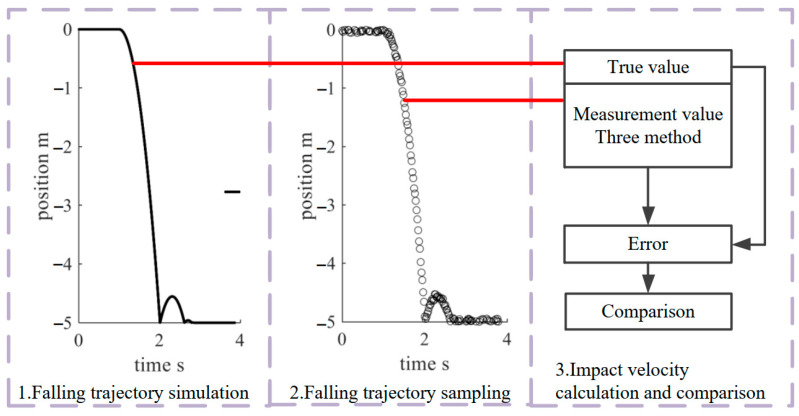
Specific steps of simulation experiment.

**Figure 5 sensors-22-08288-f005:**
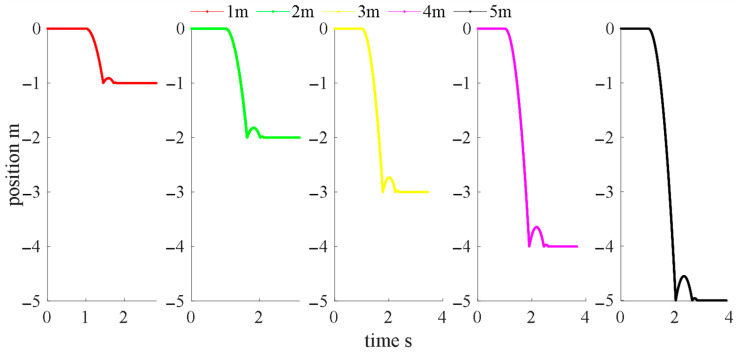
Falling trajectory simulation results.

**Figure 6 sensors-22-08288-f006:**
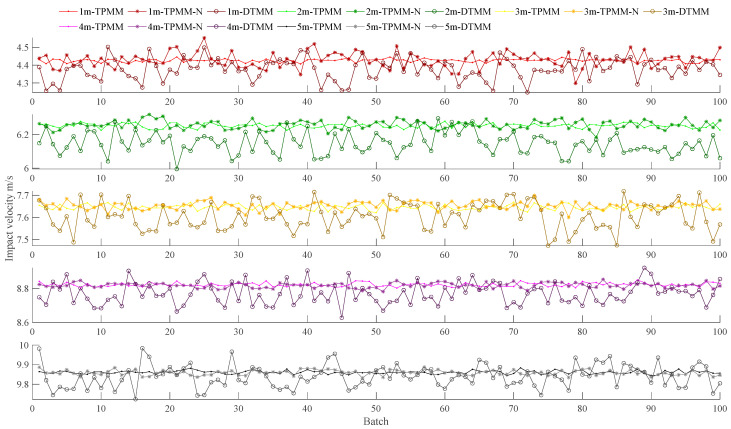
Simulation measurement results.

**Figure 7 sensors-22-08288-f007:**
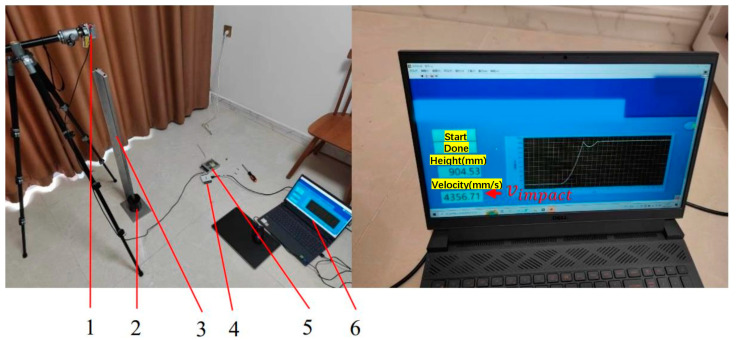
Actual experiment device. 1. Laser displacement sensor; 2. Falling weight; 3. Guide rail; 4. Data acquisition system (Ni DAQ); 5. Power supply; 6. Computer.

**Figure 8 sensors-22-08288-f008:**
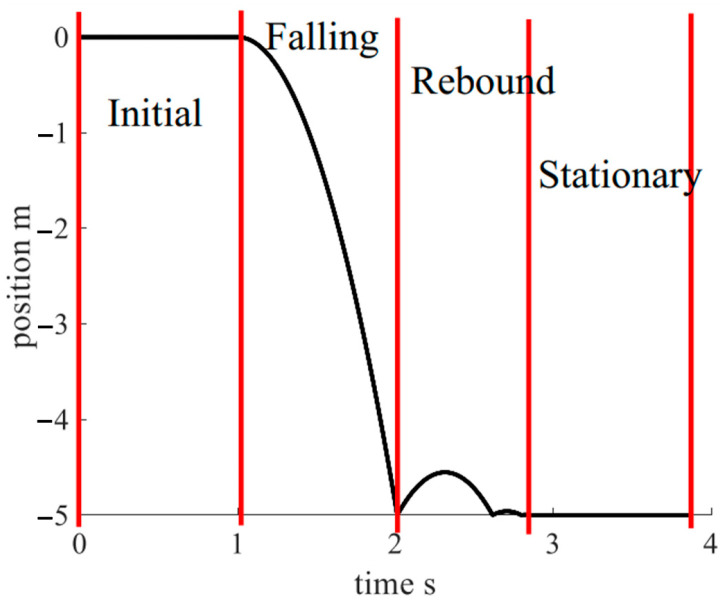
Falling process decomposition.

**Figure 9 sensors-22-08288-f009:**
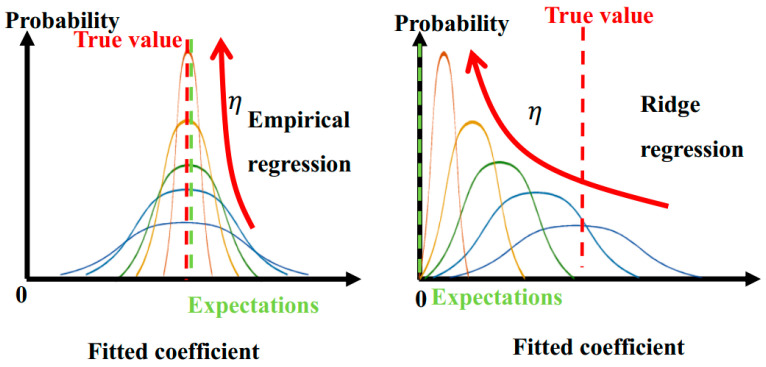
Analysis of empirical regression principle. The distribution of the fitted coefficient with different weight of the regular term (η in (6)) are shown in different colors.

**Table 1 sensors-22-08288-t001:** Values of simulation parameters.

Simulation Parameter	Value
C	0.75
ρ	$1.27\;\mathrm{kg}/m^{3}$
A	$0.01m^{2}$
M	5 kg
dt	0.0001 s
$dt_{s}$	0.01 s
Velocity decrease ratio after rebound (coefficient of restitution)	0.3
Sensor error range	±5 mm U~(−5, 5) TPMM TPMM-N
	±1 mm U~(−1, 1) DTMM

**Table 2 sensors-22-08288-t002:** Relative maximum error and relative expanded uncertainty of simulation measurement.

Height/m	Relative Maximum Error			Relative Expanded Uncertainty		
	TPMM	TPMM-N	TDMM	TPMM	TPMM-N	TDMM
1	0.481%	2.903%	4.005%	0.402%	1.936%	2.755%
2	0.470%	1.130%	4.141%	0.442%	0.818%	2.056%
3	0.462%	0.653%	2.406%	0.322%	0.470%	1.616%
4	0.409%	0.636%	2.386%	0.257%	0.338%	1.450%
5	0.335%	0.453%	1.547%	0.175%	0.283%	1.191%

The error and uncertainty of the TPMM is the lowest among all methods involved in comparison.

**Table 3 sensors-22-08288-t003:** Parameter values of the actual experiment.

Parameter	Value
Laser displacement sensor model	LR−TB5000
Measurement accuracy	±5 mm
Sampling rate	100 Hz
Falling weight outer diameter/height/inner diameter	90/95/22 mm
Falling weight mass	2.2 kg
Falling height	905 mm

**Table 4 sensors-22-08288-t004:** Results of actual impact velocity measurement experiments.

Batch	$v_{impact}\;m/s$		
	TPMM	TPMM-N	DTMM
1	4.374	4.309	4.928
2	4.381	4.369	4.751
3	4.383	4.332	5.112
4	4.390	4.406	4.562
5	4.375	4.388	5.031
6	4.380	4.283	4.713
7	4.392	4.374	5.286
8	4.385	4.388	4.840
9	4.383	4.354	4.649
10	4.398	4.345	5.095
Relative expanded uncertainty	0.342%	1.731%	10.638%

## Data Availability

The portion of the data presented in this study are available on request from the corresponding author. The raw data cannot be shared at this time as the data also forms part of an ongoing study.

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
