# Peer review of "Impact Velocity Measurement Method Based on Trajectory and Impact Position"

_sensors, 2022, doi:10.3390/s22218288_

Round 1

Reviewer 1 Report

This manuscript researches an impact velocity measurement method based on trajectory and impact position. Some problems should be considered as follows:

1. The authors applied a third-order polynomial to fit the falling trajectory. It is a commonly used fitting method. Thus the innovation or technology gap of this work should be explained more clearly.

2. In simulation experiments, the uniformly distributed random error is added into the sampling data. The rationality should be description. Further, the measurement error and measurement uncertainty of simulation measurement results are caused by the magnitude of the added random error. It is independent of the fitting method. Thus, the simulation experiments is not enough to prove the effectiveness of the method.

3. Lack of comparative experiments to illustrate the superiority.

Reviewer 2 Report

Figure 1 can be more explanatory. The relationship between the sensor positioning and colored segments of the cylinder is not stated or clear.    The authors simulate whole dynamics with the coefficient of restitution of 0.3 ( table 1 on page 7, line 190) but only measure and analyze the first pre-impact dropping. Are the authors able to measure rebound velocity with the proposed method?   To my understanding, the authors conducted a single experiment for different drop heights and generated 100 fictional data by adding uniformly distributed noise. Later, they calculated the relative error between measurements of actual data and the generated data. This can be stated more clearly. Section 3.1 can be revised.   In Table 3 (Page 9, line 222), the authors state the actual sensor accuracy is 4mm, but the accuracy range is 5mm in Table 1 (page 7, line 190)  with added uniformly distributed noise. The characteristic of added noise should be clearly stated. ( U~(-1,1) or N~(0, sigma^2) etc.)

Round 2

Reviewer 1 Report

This manuscript has been improved.

Reviewer 2 Report

The authors have taken into consideration the comments by the reviewers, and the text and the figures have been revised as per their recommendations.

Accept the paper.